# Religiosity and Beliefs toward COVID-19 Vaccination among Adults in Puerto Rico

**DOI:** 10.3390/ijerph191811729

**Published:** 2022-09-17

**Authors:** Andrea López-Cepero, McClaren Rodríguez, Veronica Joseph, Shakira F. Suglia, Vivian Colón-López, Yiana G. Toro-Garay, María D. Archevald-Cansobre, Emma Fernández-Repollet, Cynthia M. Pérez

**Affiliations:** 1Department of Epidemiology, Rollins School of Public Health, Emory University, Atlanta, GA 30322, USA; 2Department of Epidemiology, Graduate School of Public Health, University of Pittsburg, Pittsburg, PA 15261, USA; 3Department of Behavioral, Social and Health Education Sciences, Rollins School of Public Health, Emory University, Atlanta, GA 30322, USA; 4Comprehensive Cancer Center, University of Puerto Rico, San Juan 00792, Puerto Rico; 5Department of Biostatistics and Epidemiology, Graduate School of Public Health, University of Puerto Rico-Medical Sciences Campus, San Juan 00921, Puerto Rico; 6Center for Collaborative Research in Health Disparities, University of Puerto Rico-Medical Sciences Campus, San Juan 00921, Puerto Rico

**Keywords:** religiosity, COVID-19 vaccination, COVID-19, health belief model

## Abstract

Religiosity may influence COVID-19 vaccination. However, it remains unclear how religiosity is associated with beliefs toward COVID-19 and vaccination against it, particularly amongst ethnic minorities. This study examined the association between religiosity, vaccination intent, beliefs, and attitudes related to COVID-19 and vaccination among adults in Puerto Rico. Data from an online survey conducted between December 2020–February 2021 among adults (≥18 yr; *n* = 1895) residing in Puerto Rico were used. Rating of the importance of religiosity was used to capture the level of religiosity (‘less important’, ‘somewhat important’, ‘important’, and ‘very important’). The health belief model (HBM) assessed beliefs and attitudes toward COVID-19 and vaccination against it. Adjusted Poisson models with robust error variance estimated prevalence ratios (PR) and 95% confidence intervals for vaccination intent and individual COVID-19 HBM constructs. Compared to individuals rating religiosity as ‘less important’ to them, those rating it as ‘very important’ were more likely to be unwilling or uncertain to get the COVID-19 vaccine (PR = 1.51, 95% CI = 1.10–2.05). Higher ratings of importance of religiosity, compared to the lowest level, were associated with significantly lower perceived COVID-19 susceptibility, more vaccine barriers, and lower vaccine benefits (all *p* < 0.05). Individuals who reported religiosity being very important to them were more likely to report that they will get the COVID-19 vaccine only if given adequate information about it (PR = 1.14, 95% CI = 1.02–1.27) and more likely to get the vaccine if more people decide to receive it (all *p* < 0.05). In conclusion, our results suggest an association between religiosity and COVID-19 vaccination intent and beliefs and attitudes toward vaccination. The study highlights important guidelines for public health campaigns to increase vaccine uptake among religious communities in Puerto Rico.

## 1. Introduction

In the long-standing battle against COVID-19, misconceptions about COVID-19 vaccines must be addressed to reduce COVID-19 incidence and mortality [1,2,3]. In Puerto Rico, a US territory primarily comprised of Hispanic individuals, vaccines against COVID-19 became available to a select group of the population (i.e., health professionals) on 15 December 2020 [4]. Despite the island’s high vaccination rate against COVID-19 [5], intermittent outbreaks in population subgroups have occurred. This highlights the need for research evaluating factors that may have hindered timely vaccination. This is also relevant given that the Centers for Disease Control and Prevention (CDC) recommends that individuals 5 years or older get a booster vaccine amid the threat of new COVID-19 variants [6].

Religiosity has been recognized as an important predictor of health-related decision-making, and a few studies have evaluated its association with COVID-19 vaccination intent [7,8]. For instance, among adults in the US (*n* = 501; 8% Hispanic), Olagoke et al. found that a greater level of religiosity was linked with lower levels of vaccination intent [8]. Other studies in the United Kingdom (*n* = 2025) and Ireland (*n* = 1041) also report similar findings [7]. This evidence is of particular importance to individuals in Puerto Rico, given that 92% of the population reports having a religious affiliation, a statistic that is higher than that for US mainland Hispanics (83%) and the overall US mainland population (80%) [9,10]. Religion in Puerto Rico has been uniquely shaped by the island’s colonial history and is an important aspect of its culture [11,12]. Reports indicate that the majority of individuals on the island self-identify as Christian (97%), with 56% being Catholic and 33% Protestant [9]. In each of the 78 municipalities of Puerto Rico, a church is the center point around which the town’s plaza was built. Spiritually, it has been reported that a substantial proportion of individuals in Puerto Rico report religious commitment (i.e., praying daily, attending services, and it being important in their lives) [11,13]. This is of particular importance given that a previous study conducted on the island showed that, in bivariate analysis, a lower proportion of individuals rating religiosity as ‘very important’ reported an intent to get vaccinated against COVID-19 (79.5%) compared to those rating religiosity as less important (86.8%). A COVID-19 outbreak among churchgoers in Puerto Rico further highlights the need of considering religiosity in public health campaigns. This outbreak resulted in 23 cases, 12 hospitalizations, and four deaths caused by COVID-19, including a religious leader [14]. Hispanic religious communities in New York have also experienced a high number of deaths [15].

Understanding attitudes and beliefs that may explain the association between religiosity and intent is imperative to inform interventions enhancing uptake of vaccines and boosters in population subgroups in Puerto Rico. The health belief model (HBM) is a framework comprised of several constructs used to predict and understand health behaviors, including COVID-19 vaccination [16,17]. The model’s constructs include perceived susceptibility, perceived severity, perceived benefits, perceived barriers, and cues to action [18]. Understanding how religiosity is associated with HBM constructs in the context of COVID-19 vaccination may help create effective vaccination campaigns. Previous studies, including one in Puerto Rico, have shown that HBM constructs including lower perceived COVID-19 susceptibility and severity, less perceived vaccination benefits, and more barriers contribute to lower vaccination intent [17,19]. Additionally, these studies documented that specific cues to action significantly impacted vaccination intent [17,19]. Thus, the present study sought to answer the following research questions: (1) is the importance of religiosity associated with COVID-19 vaccination intent among adults in Puerto Rico? and (2) is the importance of religiosity associated with beliefs and attitudes toward COVID-19 vaccination in this population?

## 2. Materials and Methods

### 2.1. Study Design and Participants

The present study was a secondary analysis that used data from a previously described cross-sectional study [19]. In summary, an anonymous web-based questionnaire was conducted in Puerto Rico and collected data through a convenience sampling approach. The survey was created using Google Forms and distributed using social network pages (i.e., Instagram, Facebook, and Twitter) and online organizations, institutional newsletters, and academic groups as dissemination strategies. An information sheet describing the study and the questionnaire was used, given that the survey was anonymous. The online questionnaire was made available on 15 December 2020 and closed on 15 February 2021 (before vaccines were available to the general population). The Institutional Review Board at the University of Puerto Rico-Medical Sciences Campus approved the study.

To be eligible for the study, respondents had to be 18 years or older, reside on the island, have access to an electronic device, and complete the questionnaire in Spanish. A total of 1945 individuals answered the online survey and met the eligibility criteria. Individuals who reported being vaccinated against COVID-19 (*n* = 34) and those with missing data on sex at birth (*n* = 16) were excluded, resulting in a final analytic sample of 1895 surveys.

### 2.2. Study Measures

The survey was designed to collect data on the intention to get vaccinated against COVID-19, perceptions and beliefs about COVID-19 infection, health literacy, and socio-demographic, behavioral, and clinical characteristics.

#### 2.2.1. Importance of Religiosity

The importance of religiosity was measured with one item that asked, “How important are religion and religious beliefs to you?” [20]. Possible response options included less important, somewhat important, important, and very important. The ‘less important’ category was used as the reference group for analyses.

#### 2.2.2. HBM Constructs and Vaccination Intent

HBM constructs were used to measure beliefs about COVID-19 infection and vaccination, as previously done by Wong et al. [17]. To assess these constructs, survey questions asked about perceived COVID-19 susceptibility (i.e., getting COVID-19 is currently a possibility for me), severity (i.e., I will be very sick if I get COVID-19), barriers to COVID-19 vaccination (i.e., I am concerned about the safety of the COVID-19 vaccination), perceived benefits of vaccination (i.e., vaccination is a good idea because it makes me feel less worried about getting COVID-19), and cues to actions for vaccination (i.e., I will only take the COVID-19 vaccine if the vaccine is taken by many in the public). As in Wong et al., response options to these items were “Agree” or “Disagree” [17].

COVID-19 vaccination intent was assessed by asking participants, “When a vaccine for the coronavirus becomes available, will you get vaccinated?” [21]. Possible response options were “Yes”, “No”, and “Unsure”, following similar studies [21]. Concerning COVID-19 vaccine availability at the time, a fourth option stated, “I already got the vaccine.” Response options were collapsed into “Yes” and “Unsure/No”, using the “Yes” group as the reference category for analyses.

#### 2.2.3. Covariates

Socio-demographic covariates used in this analysis were obtained from the literature and included sex, age, education, employment status, annual household income, and marital status. An extensive list of health conditions, including respiratory, psychiatric, endocrine, cardiovascular, cancer, autoimmune, rheumatic, neurological, kidney, and liver diseases, was used to measure the medical history of chronic disease. A dichotomous variable was created to measure any chronic disease vs. none. Health literacy was assessed with an item that asked about the individual’s confidence in filling out medical forms by themselves, with response options ranging from extremely uncomfortable to extremely comfortable [22,23]. Perceived health was measured by asking the participants to rate their health status, with options ranging from poor to excellent.

### 2.3. Statistical Methods

Baseline characteristics were cross-tabulated with the importance of religion reported by study participants and summarized as numbers and percentages. Chi-square or Fisher’s exact tests were used to investigate the association between religiosity and each HBM construct. Adjusted Poisson models with robust error variance were used to estimate prevalence ratios (PR) and 95% confidence intervals (CI) for each of the outcomes (i.e., vaccination intent and each COVID-19 HBM construct). All regression models were adjusted for age, sex, education, health literacy, marital status, perceived health, and history of chronic disease. Statistical analyses were conducted using STATA for Macintosh release 16.1 (StataCorp L.P., College Station, TX, USA).

## 3. Results

Descriptive characteristics of the study sample by the importance of religiosity are shown in Table 1. Overall, 21.4% of respondents reported religion as less important, 17.7% as somewhat important, 26.1% as important, and 34.7% as very important. A quarter of the sample had an annual household income >$75,000, 66% were employed, and 42% perceived their health as ‘very good’. Female participants and those who were married or living with a partner were more likely to report that religion was very important to them (*p* < 0.001). Furthermore, as age and education increased, so did the proportion of participants who reported religion as very important to them (*p* < 0.001 and *p* = 0.04, respectively). Those who reported religion as very important to them were more likely to have a history of chronic diseases (*p* = 0.004) and were very comfortable filling out medical forms (*p* < 0.001).

In bivariate analyses, individuals with higher ratings of religiosity importance (‘very important’: 19.9%; ‘important’: 16.0%; and ‘somewhat important’: 18.7%) were more likely to be uncertain about getting the COVID-19 vaccine than those rating religiosity as less important to them (12.8%; *p* = 0.018) (Table 2). Additionally, bivariate analyses of HBM constructs and religiosity revealed that participants with a higher rating of importance of religiosity were less likely to perceive that getting infected with COVID-19 was a possibility for them (66.6%) than those who reported religion being less important for them (76.6%; *p* = 0.001) (Table 2). Similar results were observed for perceived benefits of vaccination; those who described religion as very important were less likely to agree that vaccination made them feel less worried about catching COVID-19 (19.9%) than participants rating religion as less important (13.6%; *p* = 0.025). Participants who reported religion as very important to them were also more concerned about the COVID-19 vaccine’s side effects (*p* < 0.001), efficacy (*p* = 0.009), and safety (*p* < 0.001) compared to those who reported religion as less important. Lastly, as the importance of religiosity increased, the proportion of individuals who reported they would only take the vaccine if it were taken by many in the public also increased (‘less important’: 13.1%; ‘somewhat important’: 20.8%; ‘important’: 19.8%; and ‘very important’: 22.0%; *p* = 0.003). None of the perceived severity constructs were significantly associated with importance of religiosity.

We observed similar results after adjusting for covariates, as shown in Table 3. Individuals who recounted that religiosity was very important were more likely to be unwilling or uncertain to get the COVID-19 vaccine once it was made available than individuals who noted religiosity was less important to them (PR = 1.51, 95% CI = 1.10–2.05). Individuals who regarded religiosity as somewhat important and very important were also more likely not to believe that getting COVID-19 was a possibility for them than participants with the lowest level of religiosity. Participants who reported that religiosity was very important to them were more likely not to perceive that the COVID-19 vaccination was a good idea than those with lower levels of importance of religiosity (PR = 1.38, 95% CI = 1.01–1.88). Vaccine barriers were also observed as the level of importance of religiosity increased. For example, individuals rating religiosity as very important to them were more likely to be concerned about vaccine side effects (PR = 1.44, 95% CI = 1.10–1.73) and more likely to have concerns over vaccine efficacy (PR = 1.22, 95% CI = 1.03–1.43) than individuals rating religiosity as less important to them. Concerns over vaccine safety were observed for all levels of importance of religiosity compared to those with the lowest rating of importance of religiosity, with PRs increasing as the level of religiosity increased. Regarding cues to get vaccinated, individuals who reported religiosity being very important to them were more likely to report that they would get the COVID-19 vaccine if given adequate information about it, compared to those rating religiosity as less important to them (PR = 1.14%, 95% CI = 1.14–1.27). In addition, compared to individuals with the lowest level of importance of religiosity, those with higher ratings were more likely to report that they would only take the vaccine if many people decide to receive it (‘somewhat important’ PR = 1.55, 95% CI = 1.12–2.15; ‘important’ PR= 1.45, 95% CI = 1.05–1.99; and ‘very important’ PR= 1.61, 95% CI = 1.19–2.19).

## 4. Discussion

### 4.1. Summary of Findings

To our knowledge, this is the first study that evaluates the association between religiosity and COVID-19 vaccination intent and beliefs in Puerto Rico. Overall, our results showed that greater religiosity was associated with being uncertain or unwilling to get the COVID-19 vaccine, lower perceived COVID-19 susceptibility and vaccination benefits, and greater vaccination barriers. Religiosity was also related to specific cues to action: receiving adequate information about the vaccine and many people deciding to receive the vaccine.

Our finding of the importance of religiosity being associated with higher vaccination uncertainty and unwillingness is in agreement with previous studies in the US and other countries [8,24,25,26,27]. A review of COVID-19 vaccine hesitancy revealed that religiosity was one of the factors influencing vaccine refusal [28]. For example, in a sample of US adults in Washington D.C., Virginia, Maryland, and Texas (*n* = 1609; 11% Hispanics), a higher level of religiosity was associated with 21% lower odds of COVID-19 vaccine acceptance [29]. Thus, our findings suggest that religiosity may be an important factor for COVID-19 vaccination hesitancy and refusal and document this phenomenon among adults in Puerto Rico, a population with a high level of religiosity affiliation that has remained largely understudied in research.

Our results also document associations between the importance of religiosity and specific HBM constructs. For instance, adults with higher ratings of religiosity importance had lower perceived susceptibility to COVID-19 (i.e., not believing that getting COVID-19 was a possibility for them), lower perceived benefits toward vaccination (i.e., not perceiving COVID-19 vaccination as a good idea), greater perceived vaccination, and only getting the vaccine if offered adequate information about it and if taken by many people. Perceived COVID-19 severity was the only HBM construct not associated with religiosity. This may be because, in our sample, perceived COVID-19 severity was high across all items studied (i.e., COVID-19 complications being serious = 97%; being afraid of getting COVID-19 = 85%; and getting very sick if contracting COVID-19 = 63%) [19]. The results regarding religiosity being associated with most HBM constructs are of particular importance, given that these factors are essential to engage in health behaviors [18], including vaccination, and may thus explain the increased COVID-19 vaccine hesitancy observed among individuals with higher levels of religiosity.

### 4.2. Potential Mechanisms of Action

To our knowledge, this is the first study evaluating the association between religiosity and HBM constructs to understand the religiosity–vaccine hesitancy link. Other studies shed light onto several factors, potentially explaining the lower perceived susceptibility and vaccination benefits and higher vaccination barriers with increasing religiosity. Olagoke et al. found that an external locus of health control (i.e., the belief that an individual’s health is dependent on external factors, such as God) mediated the association between religiosity and vaccination intent [8]. Religion is a positive source of comfort and coping during times of uncertainty and distress [30], like the COVID-19 pandemic; however, placing fate and health with divinity may account for lower perceived susceptibility observed among individuals with higher levels of religiosity. Nonetheless, studies are needed to test this hypothesis to understand better how religiosity is linked with lower perceived susceptibility to COVID-19. In addition, misinformation and obtaining health information from informal sources (e.g., social media, family and friends, and non-science websites) has been an issue throughout the COVID-19 pandemic [31,32,33]. Studies have shown that religiosity is associated with consuming health information from informal sources, mainly social media reports and friends, family, colleagues, and neighbors [8]. This may explain the observed greater perceived barriers toward vaccination, lower perception of vaccine benefits, and the need for adequate information about the COVID-19 vaccine with increasing levels of religiosity. Misinformation and trust in what others say about COVID-19 may also explain our findings of individuals with higher levels of religiosity relying on many people taking the vaccine to get it. Lastly, the observed association between religiosity and specific COVID-19 beliefs and vaccine hesitancy may be due to individuals perceiving that science interferes in and is conflicting with their religious beliefs, thus lessening their religious status [34,35]. Nonetheless, more qualitative research studies deeply exploring these factors among individuals with high levels of religiosity in Puerto Rico are needed.

### 4.3. Public Health Implications

Our findings have important public health implications for populations experiencing COVID-19 vaccine hesitancy among religious sectors. Religious leaders have an important role in promoting public health during the COVID-19 pandemic. As highlighted in Olagoke et al., leaders need to educate religious community members on their vital role to preserve their health by using faith-based justifications and scriptures [8]. This may help decrease the external locus of control and increase perceived susceptibility to disease and engagement in preventive behaviors (such as vaccination). Religious leaders should also inform their community members about their religion’s stance toward vaccination. For example, the Vatican has expressed its positive stance toward vaccination, calling it an ‘act of love’ [36]. This may help reduce the potential misbelief in science, lessening religious status. In addition, public health campaigns should partner with faith-based organizations and religious leaders to deliver a unifying message about the COVID-19 vaccines and debunk myths commonly discussed in their communities [37]. These educational campaigns need to be paired with mass vaccination events in churches led by religious leaders. Our findings on individuals with higher levels of religiosity show they are more likely to get the vaccine if many people receive it. This information may also serve as a foundation for public health campaigns to increase COVID-19 boosters and enhance immunity, particularly relevant because of the emerging COVID-19 variants. In this challenging period of public health emergencies, there is a great need for studies testing these intervention targets and strategies in culturally tailored interventions for religious communities.

As previously mentioned, over the summer of 2021, Puerto Rico had a COVID-19 outbreak traced back to a church [14], suggesting that there was resistance to vaccination in some communities on the island despite its high vaccination rate. Additionally, even though there are island-wide vaccination mandates, religious beliefs may be an exception to those mandates. These circumstances further document a need to conduct targeted public health campaigns among religious communities in Puerto Rico to safeguard the population’s health. Lastly, the sub-optimal worldwide vaccination rate (62% fully vaccinated worldwide as of 23 November 2021 [37]), the emerging COVID-19 variants, and the need for booster shots to revamp immunity indicate that countries will be coexisting with the disease for years to come. Thus, considering the current and upcoming challenges against COVID-19, our study findings are relevant to continue to impact and protect communities at high risk.

### 4.4. Study Strengths and Limitations

Several limitations need to be considered when interpreting the results of this study. The present analysis relied on a study that used online recruitment strategies and data collection methods [19]. Although this allowed for safe data collection during the pandemic, it may have introduced sampling bias as most study participants were women and individuals of higher socioeconomic status, an issue commonly encountered in studies relying on online recruitment and data collection [38,39]. Thus, study findings may not be generalizable to the entire Puerto Rico population. Similarly, generalizability may be limited to the population with similar religious affiliations given that the association between religion and vaccination intent has not been found for specific religions (i.e., Islam and Buddhism), although findings are mixed [40]. Additionally, our response option on the importance of religiosity did not include ‘not important/atheist’. Although reports have documented that only 1% of the population in Puerto Rico identifies as atheist [41], future studies should include this response option. Research should also evaluate other factors closely related to religion, such as political affiliation and conservatism, to disentangle religion’s effect on vaccination intent and beliefs. Lastly, although data on the importance of religiosity were measured, this was not the focus of the primary study; thus, the present analysis was unable to evaluate other essential variables such as internal/external locus of health control, religious attendance, and COVID-19 information sources. Qualitative studies among religious leaders and community members are needed to deeply understand the relationship between religiosity and beliefs toward COVID-19 and vaccination against it.

Nonetheless, our study has several strengths that outweigh these limitations. It provides relevant public health information to understand the link between religiosity and vaccine hesitancy. This information can be used to design and implement public health campaigns to improve vaccination uptake and decrease misinformation in a high-risk community. Lastly, our study evaluated COVID-19 beliefs using the HBM, a commonly used framework in the health behavior field [18], including COVID-19 vaccination [16,17,42].

## 5. Conclusions

In conclusion, the present study documented associations between religiosity and COVID-19 vaccination intent and specific beliefs toward the disease and vaccination against it among adults in Puerto Rico. It highlights the needed guidelines for implementing targeted public health campaigns to increase vaccine uptake among religious communities on the island.

## Figures and Tables

**Table 1 ijerph-19-11729-t001:** Sample characteristics by importance of religiosity among adults living in Puerto Rico (*n* = 1895).

Characteristic	Importance of Religiosity				
	Less Important *n* = 406 (21.4%)	Somewhat Important *n* = 336 (17.7%)	Important *n* = 495 (26.1%)	Very Important *n* = 658 (34.7%)	*p*-Value
**Female**	276 (68.0)	253 (75.3)	389 (78.6)	526 (79.9)	**<0.001**
**Age group (years)**					**<0.001**
18–29	173 (42.6)	109 (32.5)	104 (21.0)	92 (14.0)	
30–39	85 (20.9)	67 (19.9)	96 (19.4)	110 (16.7)	
40–49	84 (20.7)	67 (19.9)	116 (23.4)	157 (23.9)	
≥50	64 (15.8)	93 (27.7)	179 (36.2)	299 (45.4)	
**Highest education level**					**0.004**
High school graduate or less	15 (3.7)	14 (4.2)	22 (4.3)	19 (2.9)	
Associate degree	53 (13.0)	38 (11.3)	38 (7.7)	34 (5.2)	
Some college	23 (5.7)	13 (3.9)	23 (4.7)	42 (6.4)	
Undergraduate degree	142 (35.0)	108 (32.1)	179 (36.2)	254 (38.6)	
Master’s degree	99 (24.4)	97 (28.9)	149 (30.1)	191 (29.0)	
Doctoral degree	74 (18.2)	66 (19.6)	84 (17.0)	118 (17.9)	
**Employed**	261 (64.3)	210 (62.5)	327 (66.1)	449 (68.2)	0.285
**Annual household income**					0.237
≤$20,000	80 (19.7)	57 (17.0)	81 (16.4)	105 (16.0)	
$20,001–$40,000	83 (20.4)	84 (25.0)	126 (25.5)	179 (27.1)	
$40,001–$75,000	107 (26.4)	77 (22.9)	120 (24.1)	153 (23.3)	
>$75,000	108 (26.6)	77 (22.9)	123 (24.9)	157 (23.9)	
Prefer not to answer	28 (6.9)	41 (12.2)	45 (9.1)	64 (9.7)	
**Married/living with partner**	174 (42.9)	174 (51.8)	293 (59.2)	421 (64.0)	**<0.001**
**History of comorbidities**	294 (72.4)	240 (71.4)	361 (72.9)	526 (79.9)	**0.004**
**Confidence filling out medical forms**					**<0.001**
Extremely/a little uncomfortable	31 (7.6)	23 (6.9)	14 (2.8)	30 (4.6)	
Neutral	75 (18.5)	82 (24.3)	111 (22.5)	131 (19.9)	
Very comfortable	92 (22.7)	87 (25.9)	158 (31.9)	204 (31.0)	
Extremely comfortable	208 (51.2)	144 (42.9)	212 (42.8)	293 (44.5)	
**Perceived health**					0.070
Poor/regular	40 (9.8)	33 (9.8)	38 (7.7)	67 (10.2)	
Good	94 (23.2)	73 (21.7)	149 (30.1)	165 (25.1)	
Very good	180 (44.3)	148 (44.1)	208 (42.0)	255 (38.7)	
Excellent	92 (22.7)	82 (24.4)	100 (20.2)	171 (26.0)	

Column percentages are shown.

**Table 2 ijerph-19-11729-t002:** Bivariate association between vaccine hesitancy predictors and importance of religiosity among adults in Puerto Rico.

Characteristic	Importance of Religiosity				
	Less Important	Somewhat Important	Important	Very Important	*p*-Value
**Vaccine intent**					
Do you have plans to get vaccinated against COVID-19 when a vaccine is available?					**0.018**
Yes	354 (87.2)	273 (81.3)	416 (84.0)	527 (80.1)	
No/Unsure	52 (12.8)	63 (18.7)	79 (16.0)	131 (19.9)	
**Perceived susceptibility**					
My chance of getting COVID-19 in the next few months is high.					0.975
Agree	230 (56.6)	196 (58.3)	284 (57.4)	378 (57.5)	
Disagree	176 (43.4)	140 (41.7)	211 (42.6)	280 (42.5)	
I am worried about the likelihood of getting COVID-19.					0.125
Agree	378 (93.1)	309 (92.0)	474 (95.8)	613 (93.2)	
Disagree	28 (6.9)	27 (8.0)	21 (4.2)	45 (6.8)	
Getting COVID-19 is currently a possibility for me.					**0.001**
Agree	311 (76.6)	216 (64.3)	344 (69.5)	438 (66.6)	
Disagree	95 (23.4)	120 (35.7)	151 (30.5)	220 (33.4)	
**Perceived severity**					
The complications from contracting COVID-19 are serious.					0.949
Agree	395 (97.3)	325 (96.7)	482 (97.4)	640 (97.3)	
Disagree	11 (2.7)	11 (3.3)	13 (2.6)	18 (2.7)	
I will be very sick if I get COVID-19.					0.130
Agree	241 (59.4)	207 (61.6)	325 (65.7)	431 (65.5)	
Disagree	165 (40.6)	129 (38.4)	170 (34.3)	227 (34.5)	
I am afraid of getting COVID-19.					0.108
Agree	337 (83.0)	280 (83.3)	436 (88.1)	566 (86.0)	
Disagree	69 (17.0)	56 (16.7)	59 (11.9)	92 (14.0)	
**Perceived benefits**					
Vaccination is a good idea because it makes me feel less worried about catching COVID-19.					**0.025**
Agree	351 (86.4)	287 (85.4)	418 (84.4)	527 (80.1)	
Disagree	55 (13.6)	49 (14.6)	77 (15.6)	131 (19.9)	
Vaccination decreases my chances of getting COVID-19 or its complications.					0.121
Agree	371 (91.4)	301 (89.6)	435 (87.9)	571 (86.8)	
Disagree	35 (8.6)	35 (10.4)	60 (12.1)	87 (13.2)	
**Perceived barriers**					
I worry the possible side effects of the COVID-19 vaccination would interfere with my usual activities.					**<0.001**
Agree	116 (28.6)	123 (36.6)	192 (38.8)	291 (44.2)	
Disagree	290 (71.4)	213 (63.4)	303 (61.2)	367 (55.8)	
I am concerned about the efficacy of the COVID-19 vaccination.					**0.009**
Agree	143 (35.2)	145 (43.2)	208 (42.0)	301 (45.7)	
Disagree	263 (64.8)	191 (56.8)	287 (58.0)	357 (54.3)	
I am concerned about the safety of the COVID-19 vaccination.					**<0.001**
Agree	119 (29.3)	128 (38.1)	206 (41.6)	309 (47.0)	
Disagree	287 (70.7)	208 (61.9)	289 (58.4)	349 (53.0)	
**Cues to action**					
I will only take the COVID-19 vaccine if I am given adequate information about it.					0.135
Agree	230 (56.6)	203 (60.4)	298 (60.2)	420 (63.8)	
Disagree	176 (43.4)	133 (39.6)	197 (39.8)	238 (36.2)	
I will only take the COVID-19 vaccine if the vaccine is taken by many in the public.					**0.003**
Agree	53 (13.1)	70 (20.8)	98 (19.8)	145 (22.0)	
Disagree	353 (86.9)	266 (79.2)	397 (80.2)	513 (78.0)	

Column percentages are shown.

**Table 3 ijerph-19-11729-t003:** COVID-19 vaccine hesitancy and predictors by importance of religion using logistic regression *.

	Somewhat Important	Important	Very Important
	Adjusted PR (95% CI)	Adjusted PR (95% CI)	Adjusted PR (95% CI)
**Vaccine Intent**			
Do you have plans to get vaccinated against COVID-19 when a vaccine is available?			
Yes	1.00	1.00	1.00
No/Unsure	1.39 (0.99–1.95)	1.18 (0.84–1.64)	1.51 (1.10–2.05)
**Perceived Susceptibility**			
My chance of getting COVID-19 in the next few months is high.			
Agree	1.00	1.00	1.00
Disagree	0.95 (0.80–1.12)	0.98 (0.84–1.15)	1.00 (0.86–1.16)
I am worried about the likelihood of getting COVID-19.			
Agree	1.00	1.00	1.00
Disagree	1.21 (0.73–2.00)	0.69 (0.40–1.21)	1.13 (0.71–1.81)
Getting COVID-19 is currently a possibility for me.			
Agree	1.00	1.00	1.00
Disagree	1.46 (1.16–1.83)	1.24 (0.99–1.55)	1.36 (1.10–1.69)
Perceived Severity			
The complications from contracting COVID-19 are serious.			
Agree	1.00	1.00	1.00
Disagree	1.31 (0.59–2.90)	1.04 (0.48–2.22)	1.07 (0.50–2.26)
I will be very sick if I get COVID-19.			
Agree	1.00	1.00	1.00
Disagree	0.97 (0.82–1.15)	0.90 (0.76–1.07)	0.94 (0.80–1.11)
I am afraid of getting COVID-19.			
Agree	1.00	1.00	1.00
Disagree	1.02 (0.74–1.41)	0.75 (0.54–1.04)	0.88 (0.65–1.18)
**Perceived Benefits**			
Vaccination is a good idea because it makes me feel less worried about catching COVID-19.			
Agree	1.00	1.00	1.00
Disagree	1.03 (0.72–1.48)	1.06 (0.76–1.48)	1.38 (1.01–1.88)
Vaccination decreases my chances of getting COVID-19 or its complications.			
Agree	1.00	1.00	1.00
Disagree	1.13 (0.73–1.76)	1.29 (0.85–1.96)	1.40 (0.95–2.08)
**Perceived Barriers**			
I worry the possible side effects of the COVID-19 vaccination would interfere with my usual activities.			
Agree	1.23 (0.99–1.52)	1.29 (1.06–1.57)	1.44 (1.20–1.73)
Disagree	1.00	1.00	1.00
I am concerned about the safety of the COVID-19 vaccination.			
Agree	1.25 (1.02–1.52)	1.35 (1.12–1.63)	1.52 (1.27–1.81)
Disagree	1.00	1.00	1.00
I am concerned about the efficacy of the COVID-19 vaccination.			
Agree	1.17 (0.97–1.39)	1.12 (0.95–1.33)	1.22 (1.03–1.43)
Disagree	1.00	1.00	1.00
**Cues to Action**			
I will only take the COVID-19 vaccine if I was given adequate information about it.			
Agree	1.07 (0.94–1.20)	1.07 (0.96–1.20)	1.14 (1.02–1.27)
Disagree	1.00	1.00	1.00
I will only take the COVID-19 vaccine if the vaccine is taken by many in the public.			
Agree	1.55 (1.12–2.15)	1.45 (1.05–1.99)	1.61 (1.19–2.19)
Disagree	1.00	1.00	1.00

* The lowest level of religiosity, ‘less important’, was used as the reference category. The model was adjusted for age, sex, education, health literacy, perceived health, marital status, and history of comorbidities.

## Data Availability

Derived data supporting the findings of this study are available from the corresponding author upon request.

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
