# Peer review of "Religiosity and Beliefs toward COVID-19 Vaccination among Adults in Puerto Rico"

_ijerph, 2022, doi:10.3390/ijerph191811729_

Round 1
Reviewer 1 Report
The paper is well written and overall gives a frank account of what can and what cannot be concluded from the data at hand.
There are some major issues however that prevent me from recommending publication.
1. The value added compared to the Human Vaccines & Immunotherapeutics paper seems limited.
2. Theoretical basis is quite weak. Can we expect similar associations in other societies? Other religions? The locus of control story sounds fine but how does it explain greater concerns about safety/side effects among religious responders?
3. The data does not allow disentangling the effect of religion from effects of other variables that are correlated with religiousness and have been shown to correlate with attitudes toward Covid-19 vaccination, notably political conservatism.
4. Similarly, the data do not allow telling if vaccination attitude is a special case here or are the most religious inhabitants of Peurto Rico also hesitant to seek medical assistance in other situations.
5. Even if religiousness contributes to vaccine hesitancy if we controlled for other factors (which we cannot do, using data at hand), what would be the consequences? There is no attempt to assess efficacy of targetted pro-vax campaigns here.
Author Response
We appreciate the review of our manuscript “Religiosity and beliefs towards COVID-19 and vaccination against it in Puerto Rico”, and are submitting a revised version that addresses concerns raised by reviewers. In addition, we have highlighted in the manuscript the changes that we made. We trust that the revisions have improved the manuscript sufficiently to be considered for publication. We will gladly continue working with the reviewers and editors toward this goal.
Reviewer #1:
- The value added compared to the Human Vaccines & Immunotherapeutics paper seems limited.
Response: Understand the reviewer’s comment. However, we believe that the present manuscript makes additional important contributions given that it identifies another factor influencing vaccination intent and that affects the beliefs evaluated in the Lopez-Cepero et. al 2021 publication. As stated in the background, religion plays an important value in the Puerto Rican culture and thus we believe that our findings can inform future efforts for COVID-19 boosters and other emerging public health problems (e.g., monkeypox) in this largely understudied and underrepresented population. We have addressed the comments raised by the reviewers and in doing so we believe that the manuscript now reflects the importance of this work.
- Theoretical basis is quite weak. Can we expect similar associations in other societies? Other religions? The locus of control story sounds fine but how does it explain greater concerns about safety/side effects among religious responders?
Response: Two other explanations are now provided in addition to locus of control, which include misinformation and misbeliefs on science lessening religious status (lines 265-277). Additionally, we have added the following statement in the limitations section to address limited generalizability to other societies (line 322): Similarly, generalizability may be limited to population with similar religious affiliations to Puerto Rico given that the association between religion and vaccination intent has not been found for specific religions (i.e., Islam, Buddhism), although findings are mixed.
- The data does not allow disentangling the effect of religion from effects of other variables that are correlated with religiousness and have been shown to correlate with attitudes toward Covid-19 vaccination, notably political conservatism.
Response: We agree with the reviewer and have added this as a study limitation (line 328).
- Similarly, the data do not allow telling if vaccination attitude is a special case here or are the most religious inhabitants of Peurto Rico also hesitant to seek medical assistance in other situations.
Response: We understand and agree with the reviewer’s comment. Nonetheless, there is currently no data available from the present study or from those existing/published ones to understand vaccination behavior among religious inhabitants of Puerto Rico and address this limitation.
- Even if religiousness contributes to vaccine hesitancy if we controlled for other factors (which we cannot do, using data at hand), what would be the consequences? There is no attempt to assess efficacy of targetted pro-vax campaigns here.
Response: We acknowledge the reviewer’s comment and understand the need for such research during these challenging public health period. Because of this, lines 281-301 of the discussion highlight important intervention targets and strategies that need to be tested in future intervention studies.
Reviewer 2 Report
This is a really interesting paper that is well designed and written. It will be interesting for vaccinology and public health because the connection between religiosity and vaccination intentions is not yet described enough in current literature. However, it needs some changes before publication.
1. It would be interesting to write more about previous papers analyzing the connection between vaccination intentions and religiosity in the Introduction.
2. You should write your research questions.
3. It would be interesting to see a short description comparing the religiosity statistics in Puerto Rico vs. the United States
4. You can add a separate "Conclusions" section
5. Why do you not include an option "not important / atheist" in the question about the importance of religion?
6. Did all respondents profess the same religion?
7. I do not feel the differences between levels of importance of religion. Could you write something more about how I should interpret these levels? For example, maybe people for who religion is "very important" often attend a church than those who see religion as only "important"?
8. Try to split the Discussion into subsections, yet it isn't easy to read
9. Could you write more about which social media platforms were used to promote the survey? It is important when we take into account different attitudes toward vaccination on different social media platforms (https://www.mdpi.com/2076-393X/10/8/1190)
Author Response
We appreciate the review of our manuscript “Religiosity and beliefs towards COVID-19 and vaccination against it in Puerto Rico”, and are submitting a revised version that addresses concerns raised by reviewers. In addition, we have highlighted in the manuscript the changes that we made. We trust that the revisions have improved the manuscript sufficiently to be considered for publication. We will gladly continue working with the reviewers and editors toward this goal.
- It would be interesting to write more about previous papers analyzing the connection between vaccination intentions and religiosity in the Introduction.
Response: A greater description of the studies on religiosity and vaccination intent was added in the Introduction (line 50-57).
- You should write your research questions.
Response: The study’s research questions were added in line 84.
- It would be interesting to see a short description comparing the religiosity statistics in Puerto Rico vs. the United States
Response: A description of reports of religious affiliation among individuals in Puerto Rico, US mainland Hispanics, and the general US mainland population has been added in line 55.
- You can add a separate "Conclusions" section
Response: We have added a “Conclusions” headline section in line 344 as suggested.
- Why do you not include an option "not important / atheist" in the question about the importance of religion?
Response: We acknowledge the reviewer’s comment. Data from reports on religiosity in Puerto Rico have documented that only 1% of the population in Puerto Rico is atheist. However, we have included as a limitation in the discussion (line 325).
- Did all respondents profess the same religion?
Response: Other reports from the Puerto Rico population have documented that the majority (97%) self-identifies as Christian, with 56% identifying as catholic and 33% as protestants. This data was added in the introduction section (line 59) for context.
- I do not feel the differences between levels of importance of religion. Could you write something more about how I should interpret these levels? For example, maybe people for who religion is "very important" often attend a church than those who see religion as only "important"?
Response: We acknowledge the reviewer's comment. However, due to limitations in the available data, we cannot examine the distribution of other religion-related variables, such as service attendance, and importance of religiosity. According to the Pew Research Center, 94% of those who report religiosity being very or somewhat important to them attend religious services monthly, however, no comparison is available for other levels. We have added this as a study limitation (line 333) as future ones may evaluate attendance.
- Try to split the Discussion into subsections, yet it isn't easy to read
Response: Subsections were added in the discussion as suggested.
- Could you write more about which social media platforms were used to promote the survey? It is important when we take into account different attitudes toward vaccination on different social media platforms (https://www.mdpi.com/2076-393X/10/8/1190)
Response: The three social media platforms used to disseminate the online survey were Instagram, Facebook, and Twitter. This was added in the methods section (line 95)
Reviewer 3 Report
This is a very interesting study assessing the association between religiosity, vaccination intention, beliefs and attitudes related to COVID-19 and vaccination among adults in Puerto Rico.
In my view, the work has been carried out with methodological rigour and may be of interest to the scientific community.
One of the main limitations of the study is that the sample is not representative of the adult population of Puerto Rico (76.2% women; 82.4% with a university degree or higher). As the authors themselves point out, this means that the results cannot be extrapolated to the population as a whole.
Some changes need to be made for publication:
In the results section it is shown: “However, as the level of education increased, the importance of religiosity decreased (p=0.004)”.
This statement is not correct. Among participants with a high school degree or less, 27% considered religiosity very important, while for those with a doctorate or master's degree these percentages rose to 34.5% and 35.6%, respectively.
The results of the bivariate analysis between religiosity and intention to vaccinate (shown in Table 2) are very interesting and should be discussed in the text, in the results section.
Information from the 'Institutional Review Board Statement' and 'Informed Consent Statement' sections should also be added.
Kind regards
Author Response
We appreciate the review of our manuscript “Religiosity and beliefs towards COVID-19 and vaccination against it in Puerto Rico”, and are submitting a revised version that addresses concerns raised by reviewers. In addition, we have highlighted in the manuscript the changes that we made. We trust that the revisions have improved the manuscript sufficiently to be considered for publication. We will gladly continue working with the reviewers and editors toward this goal.
- In the results section it is shown: “However, as the level of education increased, the importance of religiosity decreased (p=0.004)”.This statement is not correct. Among participants with a high school degree or less, 27% considered religiosity very important, while for those with a doctorate or master's degree these percentages rose to 34.5% and 35.6%, respectively.
Response: The reviewer is correct. We had revised the statement to “Furthermore, as age and education increased, so did the proportion of participants who reported religion as very important to them (p<0.001 and p=0.04, respectively).”
- The results of the bivariate analysis between religiosity and intention to vaccinate (shown in Table 2) are very interesting and should be discussed in the text, in the results section.
Response: A more detailed description of the results presented in Table 2 has been added to the text (line 170).
- Information from the 'Institutional Review Board Statement' and 'Informed Consent Statement' sections should also be added.
Response: Information on IRB was added in line 99 as well as the use of an information sheet over a consent form; line 96 (An information sheet describing the study and the questionnaire was used given that the survey was anonymous and there was no study compensation due to the short duration of the survey).
Round 2
Reviewer 2 Report
The authors responded to all my comments and suggestions. The paper is yet much better structured and contains all the necessary information.
Author Response
We thank the reviewer for their valuable comments in their prior revision and are satisfied that they found the resubmission being more structured and containing all the required information. The manuscript was further revised to incorporate suggestions from the Academic Editor.